# Targeting S1PRs as a Therapeutic Strategy for Inflammatory Bone Loss Diseases—Beyond Regulating S1P Signaling

**DOI:** 10.3390/ijms22094411

**Published:** 2021-04-23

**Authors:** Hong Yu

**Affiliations:** Department of Oral Health Sciences, College of Dental Medicine, Medical University of South Carolina, Charleston, SC 29425, USA; yuho@musc.edu

**Keywords:** bone loss, sphingosine-1-phosphate receptors, cytokine, chemotaxis, osteoclastogenesis, JTE013, FTY720

## Abstract

As G protein coupled receptors, sphingosine-1-phosphate receptors (S1PRs) have recently gained attention for their role in modulating inflammatory bone loss diseases. Notably, in murine studies inhibiting S1PR2 by its specific inhibitor, JTE013, alleviated osteoporosis induced by RANKL and attenuated periodontal alveolar bone loss induced by oral bacterial inflammation. Treatment with a multiple S1PRs modulator, FTY720, also suppressed ovariectomy-induced osteoporosis, collagen or adjuvant-induced arthritis, and apical periodontitis in mice. However, most previous studies and reviews have focused mainly on how S1PRs manipulate S1P signaling pathways, subsequently affecting various diseases. In this review, we summarize the underlying mechanisms associated with JTE013 and FTY720 in modulating inflammatory cytokine release, cell chemotaxis, and osteoclastogenesis, subsequently influencing inflammatory bone loss diseases. Studies from our group and from other labs indicate that S1PRs not only control S1P signaling, they also regulate signaling pathways induced by other stimuli, including bacteria, lipopolysaccharide (LPS), bile acid, receptor activator of nuclear factor κB ligand (RANKL), IL-6, and vitamin D. JTE013 and FTY720 alleviate inflammatory bone loss by decreasing the production of inflammatory cytokines and chemokines, reducing chemotaxis of inflammatory cells from blood circulation to bone and soft tissues, and suppressing RANKL-induced osteoclast formation.

## 1. Introduction

Local or systemic bone loss occurs in many human diseases, including rheumatoid arthritis, systemic lupus erythematosus, axial spondyloarthritis, psoriatic arthritis, inflammatory bowel disease, postmenopausal osteoporosis, and periodontitis [1]. Osteoclasts, the giant multinucleated bone resorption cells, are derived from hematopoietic monocyte/macrophage lineage [2,3,4]. Typically, these cells express CD14, colony stimulating factor 1 receptor (CSF-1R), CD11b, and receptor activator of nuclear factor κB (RANK) [2,3,4]. Osteoclasts are formed by adhesion, followed by fusion of monocytes and macrophages [2,3,4]. During normal physiological conditions, most of these osteoclast precursors originate in the bone marrow and circulate in the blood. During inflammatory conditions, inflammatory cytokines and chemokines recruit monocytes from blood circulation to bone and surrounding soft tissues. Following such recruitment, the monocytes undergo differentiation, adhesion, and fusion to form giant multinucleated osteoclasts. Two essential cytokines, macrophage-colony stimulating factor (M-CSF) and RANKL, are required for the differentiation of osteoclasts [3,4,5]. M-CSF and RANKL can be generated by osteoblasts, osteocytes, bone marrow stromal cells, or T lymphocytes. M-CSF binds to its receptor CSF1R on osteoclast precursors, supporting the survival and proliferation of those osteoclast precursors [6]. In contrast, RANKL binds to its receptor RANK on osteoclast precursors, initiating cell differentiation, adhesion, and fusion, resulting in the formation of multinucleated osteoclasts [6]. It is now clear that RANKL is the indispensable and exclusive cytokine for osteoclastogenesis [3,4,5]. Another important cytokine, osteoprotegerin (OPG), is a decoy receptor for RANKL, which inhibits RANKL binding to RANK and subsequently suppresses osteoclastogenesis [3,4]. Additionally, other inflammatory cytokines, such as interleukin (IL)-1, IL-6, and tumor necrosis factor (TNF)-α, are implicated in osteoclast formation in periodontal diseases, postmenopausal osteoporosis, and rheumatoid arthritis [7,8,9]. Interestingly, these inflammatory cytokines fail to induce osteoclastogenesis in monocyte culture in the absence of RANKL, suggesting indirect mechanisms of action. Studies have revealed that these inflammatory cytokines function by promoting RANKL production, reducing OPG production, and/or up-regulating RANK on osteoclast precursors, indirectly affecting osteoclastogenesis [4,10,11]. Meanwhile, these inflammatory cytokines inhibit osteoblast proliferation [12,13] and disrupt unidirectional osteoblast alignment [14,15], resulting in improper bone regeneration. Sphingosine-1-phosphate (S1P) is a bioactive sphingolipid. S1P binds to five G protein-coupled receptors including S1PR1-S1PR5. S1PR1 selectively couples to members of the G_i_ family; S1PR2 and S1PR3 couple to members of the G_i_, G_q_, and G_12/13_ families; S1PR4 couples to G_i_ and possibly G_12/13_ families; and S1PR5 couples to G_i_ and G_12/13_ families [16,17,18]. S1P binds to S1PRs, which regulate various functions, including cell survival and growth, cell migration, and cytoskeleton organization [16,17,18,19]. In mammals, S1PR1, S1PR2, and S1PR3 are ubiquitously expressed in all tissues [16,19]. In contrast, S1PR4 is only detected in the lungs and lymphoid tissues [16,19]; also, S1PR5 is only expressed in the brain (within white matter), in skin, and in the spleen [16,19]. JTE013 is a specific antagonist of S1PR2 and FTY720 is a modulator of multiple S1PRs. In this review, we discuss recent advances in understanding the impact of S1P and S1PRs on osteoclastogenesis. In particular, we decipher the underlying mechanisms for JTE013 and FTY720 controlling inflammatory cytokine production, cell chemotaxis, and RANKL-induced osteoclastogenesis, which subsequently modulate inflammatory bone loss diseases.

## 2. Biological Effects of S1P in Inflammatory Bone Loss Diseases

### 2.1. S1P Biosynthesis and Degradation

S1P is derived from sphingosine when sphingosine kinase (SphK) 1 and/or SphK2 are activated by various stimuli, including growth factors, hormones, bacterial stimuli, and cytokines (Figure 1) [16,20,21]. SphK1 and SphK2 share an overall homology and generate the same product, S1P. However, they occupy different subcellular locations and possibly have distinct and overlapping functions [22]. SphK1 normally resides in the cytosolic compartments and can be translocated to the plasma membrane after its activation [23,24]. In contrast, SphK2 is mainly localized in the nucleus with low levels found in the cytoplasm [23,24]. SphK1 appears to be a major determinant of S1P production within the cytoplasm and membrane [25], while SphK2 mainly accounts for nuclear S1P formation [25]. We previously infected murine bone marrow derived monocytes and macrophages (BMMs) with an oral bacterial pathogen *Aggregatibacter actinomycetemcomitans* (*A. actinomycetemcomitans*). The SphK1 (but not SphK2) mRNA level significantly increased at 2, 4, or 8 h after bacterial infection [26]. This suggests that SphK1 plays an important role in promoting the generation of S1P in response to bacteria or cytokine stimuli.

It is well known that a significant variance of S1P concentrations occurs in the blood compared with S1P concentrations in the tissues [16,27,28]. Constitutive S1P levels in the tissues are very low (below 100 nM) because S1P can be degraded by S1P lyase or dephosphorylated by S1P phosphatases in the tissues [27]. In contrast, S1P is generated by vascular endothelial cells, erythrocytes, and platelets in the blood. Yet the platelets lack S1P lyase, and the erythrocytes lack both S1P lase and S1P phosphatases [16,27,28]. Hence, the serum S1P levels are very high (low μM concentration) [27]. This sharp gradient between the S1P level in the tissues and the S1P level in the blood controls the migration of immune cells from blood to peripheral tissues, subsequently influencing inflammatory responses in the tissues [16,27,28].

### 2.2. High S1P and SphK1 Levels Coincide with Inflammatory Bone Loss Diseases

Because inflammation stimulates the activation of SphK1 and the generation of S1P, high S1P and SphK1 levels are often observed in patients with inflammatory bone diseases. For example, postmenopausal women exhibit significantly higher serum S1P levels compared with the serum S1P levels in premenopausal women and in men [29]. Moreover, serum S1P level is positively correlated with bone resorption markers in postmenopausal women [29]. Patients with rheumatoid arthritis display significantly higher levels of S1P in their synovial fluid when compared with the S1P levels in non-inflammatory osteoarthritis patients [30]. In patients with apical periodontitis, the SphK1 level is also significantly increased in the periodontal tissues compared with normal tissues derived from tooth extraction [31]. Additionally, Moritz et al. (manuscript under review for publication) analyzed serum S1P levels in 3371 patients and discovered that the serum S1P concentration was positively correlated in patients with moderate and severe periodontitis, and high serum S1P level in patients with periodontitis correlated positively with the degree of periodontal pocket depth and bone attachment loss. Moreover, the genetic deletion of SphK1 in mice resulted in reduced synovial inflammation and joint erosion induced by TNF compared with wild type mice [32]. SphK1 deficiency in mice also attenuated periodontal alveolar bone loss induced by oral bacterial inflammation compared with wild type mice [26]. These results from human and murine studies suggest that S1P plays an important role in regulating inflammatory bone loss diseases.

### 2.3. Mechanisms Associated with S1P in Regulating Inflammatory Bone Loss Diseases

Inflammatory cytokine production, cell chemotaxis, and osteoclastogenesis are three important elements in the pathogenesis of inflammatory bone loss disease. Our study revealed that the addition of extracellular S1P (125 nM to 1000 nM) to murine BMMs did not induce a significant inflammatory response [26]. Instead, S1P acts as a chemokine, which promotes chemotaxis of osteoclast precursors [26]. The addition of extracellular S1P (125 nM to 1000 nM) to murine BMMs dose-dependently enhanced BMMs chemotaxis [26]. Besides S1P, other inflammatory cytokines, such as IL-1, IL-6, and TNF-α, can attract monocytes and macrophages [33,34]. The addition of both S1P and the oral bacteria (*A. actinomycetemcomitans*)-stimulated cell culture media further enhances BMMs chemotaxis [26]. Studies have revealed that S1P not only recruits monocytes, but also attracts osteoblasts and T lymphocytes [35]. Recruitment of monocytes, osteoblasts, and T lymphocytes to bone and surrounding soft tissues increases the interaction of these cells with various stimuli (such as antigens, bacteria, and cytokines), which subsequently promotes the generation of inflammatory cytokines (including RANKL). Meanwhile, the accumulation of these osteoclast precursors on the bone surfaces also enhances cell adhesion and fusion, resulting in osteoclast formation. However, the addition of extracellular S1P (1 nM to 1000 nM) to a single culture of murine BMMs did not affect RANKL-induced osteoclast differentiation [35], which suggests that S1P does not have a significant impact on the RANKL-RANK signaling pathway in BMMs. In contrast, the addition of S1P to the co-culture of both murine BMMs and osteoblasts enhanced RANKL-induced osteoclastogenesis [35], which was associated with S1P promoting osteoblast chemotaxis and enhancing the survival of osteoblasts, subsequently increasing RANKL production [35,36]. Interestingly, a previous study [37] revealed that the addition of extracellular S1P (0.5 to 1.0 µM) to murine bone marrow cells not only increased RANKL, but also enhanced OPG generation; also, the ratio of OPG/RANKL was increased in osteoblasts. These results demonstrated that S1P functions in both osteoclastogenesis and osteogenesis. In vivo studies have also showed that S1P lyase-deficient mice or pharmacological inhibition of S1P lyase, which increases S1P level, markedly enhanced bone mass and strength in mice [38]. Findings from these studies indicate that during normal physiological conditions (without inflammatory cytokine release), S1P acts mainly to promote osteoblast survival and proliferation, resulting in enhanced bone mass and strength in animals. During inflammatory conditions, S1P is generated with other inflammatory cytokines, which synergistically promote the recruitment of osteoclast precursors to the bone surface and enhance RANKL production, leading to osteoclast formation.

## 3. Biological Effects of S1PR2 in Inflammatory Bone Loss Diseases

### 3.1. Inhibition of S1PR2 by JTE013 Alleviated Inflammatory Bone Loss Diseases

S1PR2, also called endothelial differentiation G-protein coupled receptor 5 (EDG5), is located on the plasma membrane and in the cytoplasm of mammalian cells [16]. S1PR2 couples with heterotrimeric G_i_, G_q_, G_12/13_ proteins, which regulates various cellular signaling pathways, including adenylate cyclase, phospholipase C, (PLC), phosphoinositide-3 kinase (PI3K), nuclear kappa-B (NF-κB), extracellular signal-regulated kinase (ERK), c-Jun N-terminal kinase (JNK), p38 mitogen-activated kinase (MAPK), and small G protein Rac and Rho [16,17,18,19].

The studies from our and other labs demonstrate that S1PR2 plays an essential role in modulating osteoclastogenesis and bone homeostasis. Ishii et al. [39] showed that genetic deletion of S1PR2 in mice resulted in increased bone volume, numbers of trabecular bone, and trabecular thickness compared with wild type mice. They also demonstrated that pharmacological inhibition of S1PR2 by a S1PR2 inhibitor, JTE013, attenuated osteoporosis induced by RANKL [39]. In our studies, treatment with JTE013 in mice also alleviated periodontal inflammatory bone loss induced by tooth ligature placement [40]. These experimental data indicate that S1PR2 is a good therapeutic target for treatment of inflammatory bone loss-associated diseases.

### 3.2. Mechanisms Associated with S1PR2 in Regulating Inflammatory Bone Loss Diseases

#### 3.2.1. Role of S1PR2 in Inflammatory Cytokine Release

A previous study demonstrated that the genetic deletion of S1PR2 in *Apoe*^−/−^*S1pr2*^−/−^ mice displayed reduced serum IL-1β and IL-18 levels when challenged by bacterial LPS compared to *Apoe*^−/−^ mice [41]. Pharmacological inhibition of S1PR2 by JTE013 in wild type mice also reduced serum IL-1β and IL-18 levels when challenged by LPS [41]. In our studies, knockdown of S1PR2 by a S1PR2 shRNA or pharmacological inhibition of S1PR2 by JTE013 in murine BMMs decreased IL-1β, IL-6, and TNF-α inflammatory cytokine levels that were induced by the oral bacterial pathogen *A. actinomycetemcomitans* [42,43]. Subsequently, we demonstrated that treatment with either S1PR2 shRNA or JTE013 reduced PI3K, NF-κB, ERK, JNK, and p38-MAPK signaling pathways induced by *A. actinomycetemcomitans*, compared with controls [42,43]. In a ligature-induced periodontitis animal study, oral topical administration of JTE013 significantly decreased IL-1β, IL-6, and TNF mRNA levels in gingival mucosa tissues when compared with a vehicle treatment group [40]. Additionally, in a bile duct ligation-induced cholestatic liver injury study in mice, treatment with a glucan-encapsulated S1PR2 siRNA significantly attenuated IL-1β and IL-18 in the liver, as well as serum IL-1β and IL-18 levels, compared with controls [44]. Zhao et al. [45] revealed that S1PR2 is a receptor for bile acid. Deoxycholic acid dose-dependently stimulated the up-regulation of S1PR2 [45]. Moreover, deoxycholic acid induced the generation of IL-1β in macrophages, which was blocked by inhibiting S1PR2 by JTE013 [45]. In a colitis animal study induced by deoxycholic acid and dextran sulfate sodium, treatment with JTE013 alleviated inflammation in the colon [45]. Additionally, in an ovalbumin-induced experimental asthma study, mice treated with JTE013 exhibited significantly reduced levels of IL-4, IL-5, and IL-13 in the bronchoalveolar lavage fluid, as well as attenuated inflammation in the lungs [46]. In a bleomycin-induced lung fibrosis animal study, genetic deletion of S1PR2 in mice displayed attenuated lung fibrosis compared with wild type mice [47]. Zhao et al. [47] showed that bleomycin administration stimulated the mRNA expression of profibrotic cytokines (IL-13, and IL-4), as well as chemokine C-C motif ligand 17 (CCL17) and CCL24 in the lungs, which were markedly diminished in *S1pr*^−/−^ mice compared with wild type mice. Pharmacological inhibition of S1PR2 by a S1PR2 antagonist, S1PR2i, inhibited lung fibrosis induced by bleomycin [47]. These studies demonstrated that S1PR2 plays an essential role in regulating the release of inflammatory cytokines and chemokines induced by various stimuli, including bacterial pathogens, LPS, bile acid, ovalbumin, and bleomycin. Interestingly, in a sepsis animal study, Hou et al. [48] demonstrated that S1PR2 also inhibits bacterial phagocytosis. Genetic deletion of S1PR2 or pharmacological inhibition of S1PR2 by JTE013 in mice reduced bacterial burden and improved survival rate in mice with sepsis by enhancing bacterial phagocytosis [48]. This was associated with S1PR2 in regulating Rac1 and filament actin (F-actin) in response to *E. coli* [48]. Genetic deletion of S1PR2 increased Rac1 and F-actin induced by *E. coli*, subsequently enhancing bacterial uptake and phagocytosis [48]. Because enhancing bacterial phagocytosis reduces the amount of bacteria in contact with immune cells and subsequently decreases the inflammatory response, the reduction of inflammatory cytokine observed in our studies [42,43] was associated with JTE013’s ability to promote bacterial phagocytosis and attenuate LPS-induced cytokine release.

#### 3.2.2. Role of S1PR2 in Cell Chemotaxis

Previously, Ishii et al. [39] showed that murine BMMs treated with S1PR2 siRNA increased cell migration from a low concentration of S1P toward a high concentration of S1P, indicating that S1PR2 plays a chemorepulsive role induced by S1P [39]. In a RANKL-induced osteoporosis animal study, treatment with JTE013 in mice increased the number of CD11b^+^ monocytes in the blood compared with control [39]. In contrast, there was no significant difference in the number of CD3^+^ T lymphocytes in the blood between JTE013-treated mice and control [39]. This suggested that inhibition of S1PR2 by JTE013 suppressed monocytes migration from blood to bone tissues. In contrast, Yang et al. [49] showed that murine BMMs, treated with either a S1PR2 siRNA or JTE013, inhibited cell migration induced by S1P (100 nM) [49]. Using a bile duct ligation-induced cholestatic liver injury animal model, Yang et al. [49] demonstrated that mice treated with JTE013 had alleviated inflammation and fibrosis in the liver compared with controls. Treatment with JTE013 decreased the percentage of F4/80 positive macrophages in the liver compared with vehicle treatment, which also suggested that JTE013 inhibited macrophage chemotaxis from the blood circulation to the liver [49]. Because bile duct ligation stimulates the generation of various inflammatory cytokines along with S1P, both inflammatory cytokines and S1P contribute to cell chemotaxis. Therefore, it is important to evaluate the role of S1PR2 in cell chemotaxis stimulated by inflammatory cytokines. A previous study showed that treatment with a S1PR2 siRNA reduced IL-1β and IL-18 mRNA levels in liver induced by bile duct ligation [44], and treatment with JTE013 reduced IL-1β induced by deoxycholic acid [45]. Therefore, it is possible that JTE013 reduced inflammatory cytokines in the liver induced by bile duct ligation and subsequently decreased macrophage chemotaxis. In our studies, in JTE013-treated murine BMMs, we not only observed a significant reduction of levels of IL-1β, IL-6, and TNF-α induced by the oral bacterial pathogen *A. actinomycetemcomitans*, we also noticed a significant reduction of S1P in murine BMMs treated with JTE013, with or without bacterial infection [43], which suggested that JTE013 down-regulates Sphk1 activity. Because treatment with JTE013 reduced both S1P and inflammatory cytokines, JTE013 inhibited monocytes chemotaxis induced by bacteria-stimulated cell culture media [43]. Treatment with S1PR2 shRNA also suppressed monocyte chemotaxis induced by bacteria-stimulated cell culture media [42,43]. In a bleomycin-induced lung fibrosis animal study, S1PR2 deficiency reduced the total number of cells and the number of macrophages in the bronchoalveolar lavage fluid [47]. These studies demonstrated that S1PR2 controls inflammatory cytokine and/or S1P release, subsequently affecting inflammatory cell chemotaxis. S1PR2 deficiency or pharmacological inhibition of S1PR2 by JTE013 inhibited cell chemotaxis induced by various stimuli.

#### 3.2.3. Role of S1PR2 in RANKL-Induced Osteoclastogenesis

A previous study [39] in mice showed that pharmacological inhibition of S1PR2 by JTE013 suppressed RANKL-induced osteoporosis. Our studies revealed that S1PR2 controls cell adhesion units (podosomes) induced by RANKL in BMMs, which influence both osteoclastogenesis and bone resorption. Podosomes are basic cell adhesion units that are required for cell adhesion and fusion to form multinucleated osteoclasts [50,51,52]. RANKL stimulated the up-regulation of podosome components (including PI3K, Src, Pyk2, F-actin, integrin β3, and paxillin levels), which were suppressed by treatment with either S1PR2 shRNA or JTE013 compared with controls [43]. This effect was not associated with the S1P signal, since S1P has no effect on the differentiation of osteoclasts in the single culture of BMMs [35]. Treatment with S1PR2 shRNA or JTE013 in BMMs significantly decreased various osteoclastogenic genes, including nuclear factor of activated T-cells cytoplasmic calcineurin-dependent 1 (Nfatc1), cathepsin K (Ctsk), acid phosphatase 5 (Acp5), osteoclast-associated receptor (Oscar), dendritic cell-specific transmembrane protein (Dc-stamp), and osteoclast stimulatory transmembrane protein (Oc-stamp) induced by RANKL compared with controls [42,43]. Using a ligature placement induced periodontitis animal model, we demonstrated that treatment with JTE013 reduced the number of osteoclasts in the periodontal tissues and attenuated alveolar bone loss [40]. Our study demonstrated that S1PR2 plays a key role in regulating RANKL-induced osteoclastogenesis. Treatment with JTE013 inhibits inflammatory bone loss by multiple mechanisms, including suppressing the production of inflammatory cytokines and S1P, reducing monocyte chemotaxis, and inhibiting RANKL-induced adhesion and fusion of osteoclast precursors (Figure 2).

#### 3.2.4. Possible Interaction of S1PR2 with Other Signaling Molecules in Lipid Rafts

It is well known that mammalian membrane contains specialized membrane domains, called lipid rafts, which are enriched in cholesterol, glycosphingolipid, and proteins. Lipids rafts serve as signaling platforms that recruit transmembrane and intracellular signaling molecules, facilitating the interaction of signaling molecules and signaling transduction following various stimuli [53,54]. Heterotrimeric G proteins, Src, PI3K, integrins, and MAPKs are some of the signaling molecules within the lipid rafts [53,54]. Additionally, it has been shown that toll-like receptor 4 (TLR4) and its adaptor protein MyD88 are recruited to the lipid rafts in response to oxidase stress or fatty acid treatment [55,56]. RANK and its adaptor protein TRAF6 are also recruited to lipid rafts after RANKL stimulation [57]. Therefore, it is possible that S1PR2 might interact with heterotrimeric G_i_, G_q_, G_12/13_ proteins, TLR4, MyD88, RANK, TRAF6, Src, PI3K, integrins, and MAPKs within the lipid rafts after bacterial LPS or RANKL stimulation. Interestingly, it has been reported that treatment with vitamin D and its analog, eldecalcitol, reduced S1PR2 mRNA levels in murine circulating osteoclast precursors and alleviated ovariectomy-induced osteoporosis [58]. Additionally, incubation of murine bone marrow cells with IL-6 increased S1PR2 mRNA levels [59]. Treatment with an IL-6 receptor antibody alleviated collagen-induced bone loss by reducing S1PR2 in bone marrow cells [59]. These data support our hypothesis that S1PR2, as a G-protein coupled receptor, not only modulates S1P signaling, but also controls other stimuli, including bacterial LPS, RANKL, bile acid, vitamin D, and IL-6 [42,43,48,58,59] (Table 1). It is likely that S1PR2 interacts with various signaling molecules in the lipid rafts, subsequently modulating bone homeostasis.

## 4. Biological Effects of a S1PRs Modulator, FTR720, in Inflammatory Bone Loss Diseases

### 4.1. FTY720 Attenuated Inflammatory Bone Loss in Animal Studies

FTY720, also called fingolimod, is a S1PRs modulator. FTY720 is synthesized by structural modification of myriocin, a fungal metabolite from *Isaclaria sinclarii*, a traditional herb used in Eastern medicine [60]. FTY720 is phosphorylated to p-FTY720 by SphK2. Initially, FTY720 was considered a multiple S1PRs (including S1PR1, S1PR3-5) agonist [61,62]. However, later studies discovered that FTY720 also functions as a noncompetitive inhibitor of multiple S1PRs by promoting internalization and partial degradation of S1PRs [63,64]. In previous in vitro studies, FTY720 inhibited the binding of S1P to S1PR1, S1PR5, and to a lesser extent, S1PR2 in lymphocytes [63]. Additionally, FTY720 reduced S1PR1 and S1PR4 levels in dendritic cells [64]. FTY720 has been used in clinical trials as an immune suppressant to treat patients with autoimmune diseases, including relapsing multiple sclerosis, as well as in patients with renal transplant to prevent rejection of the transplant [65]. Additionally, FTY720 exhibits an anti-inflammatory bone loss effect in animals. Treatment with FTY720 in mice alleviated ovariectomy-induced osteoporosis [66,67], attenuated apical periodontitis [31], and suppressed bone destruction and hindpaw edema in animals with arthritis induced either by collagen, adjuvant, or an arthrogenic anti-collagen II antibody [67,68,69,70,71].

### 4.2. Mechanisms Associated with FTY720 in Modulating Inflammatory Bone Loss Diseases

#### 4.2.1. Role of FTY720 in Inflammatory Cytokine Response

We previously reported that treatment with FTY720 (2 to 8 μM) in murine BMMs dose-dependently reduced IL-1β, IL-6, and TNF-α induced by the oral bacterial pathogen *A. actinomycetemcomitans* by inhibiting PI3K, ERK, and Akt signaling pathways [72]. FTY720 also suppressed IL-1β, IL-6, and TNF-α induced by LPS in microglia [73], and reduced IL-6, IL-12, TNF-α, and MCP-1 induced by LPS in bone marrow-derived dendritic cells (BMDCs) [74]. Zeng et al. [74] revealed that the anti-inflammatory effect of FTY720 was associated with FTY720 in altering cell shape, surface markers, and antigen presentation induced by LPS in BMDCs [74]. Upon LPS stimulation, the shape of BMDCs became elongated. This elongation of BMDCs was suppressed by treating BMDCs with FTY720 [74]. After stimulation by microbial products or cytokines, immature dendritic cells acquire certain surface markers (including MHC II and co-stimulatory molecules), leading to phenotypic and functional maturation processes. Zeng et al. [74] showed that LPS induced up-regulation of surface markers (including MHC II molecule, I-A^d^, co-stimulatory molecule CD86, and adhesion molecule CD40) that were suppressed by treatment with FTY720 [74]. In a collagen-induced arthritis animal model, FTY720-treated mice displayed reduced IL-1β, IL-6, and TNF-α in plasma and decreased IL-6, and TNF-α in synovial tissues [69]. Additionally, FTY720 altered cytokine profiles in dendritic cells [64]. Mature dendritic cells generate distinct Th lineage-polarizing cytokines that induce clonal expansion of naïve T cells and initiate a primary adaptive immune response. Muller et al. [64] showed that co-culturing of naïve T cells with FTY720 or p-FTY720-treated dendritic cells enhanced the production of IL-4, but reduced the production of IFN-γ, suggesting that FTY720 promoted the shift from the Th1 to Th2 cytokine profile. Moreover, high concentration (3 to 10 μM) of FTY720 can also increase cell membrane permeability and promote necrosis of inflammatory cells [75], which in turn decrease inflammatory cytokine production. Furthermore, in a collagen-induced arthritis animal study, FTY720 also reduced an anti-type II collagen antibody [70]. These in vitro and in vivo studies demonstrated FTY720′s ability to inhibit inflammatory cytokine production, alter cytokine Th profiles, and suppress antibody production.

#### 4.2.2. Role of FTY720 in Regulating Cell Migration and Chemotaxis

In animals treated with FTY720, the number of lymphocytes was markedly decreased in the peripheral blood, thoracic lymph duct, and partially in the spleen. In contrast, the number of lymphocytes in peripheral lymph nodes, mesenteric lymph nodes, and Peyer’s patch was significantly increased [76]. This effect was caused by degradation of S1PR1 by FTY720, which reduces the adhesion of T cells on the lymph node sinus and suppresses the egress of lymphocytes from the secondary lymphoid organs to lymph and peripheral blood, leading to lymphopenia [77,78,79]. In adjuvant-induced arthritis studies, FTY720 treatment (0.03 to 0.3 mg/kg) significantly reduced the number of lymphocytes in the peripheral blood, without affecting the number of leukocytes and monocytes in the blood [70,71]. In a collagen-induced arthritis study, FTY720-treated animals had reduced CD4^+^ T cell infiltration in the synovium [68]. Additionally, FTY720 controls dendritic cell migration induced by chemokines [64] and modulates dendritic cell migration from peripheral tissues to draining lymph nodes [69]. Muller et al. [64] revealed that dendritic cells treated with either FTY720 or p-FTY720 diminished dendritic cells chemotaxis in response to chemokines (RANTES or SDF-1α). This effect was associated with a reduction of F-actin after treatment with FTY720 or p-FTY72 in BMDCs [64]. Dendritic cells serve as sentinels in the immune system by capturing and processing antigens at peripheral tissues. After activation of immature dendritic cells by microbial products or inflammatory cytokines, dendritic cells undergo cell maturation and migration to draining lymph nodes where dendritic cells interact with naïve T lymphocytes and convert them into antigen-specific reactive T cells. In a collagen-induced arthritis study in mice [69], treatment with FTY720 (5 mg/kg) inhibited dendritic cell migration from peripheral tissues to draining lymph nodes. This was associated with inhibiting production of chemokine CCL19 and chemokine receptor CCR7 in dendritic cells treated with FTY720 [69]. In other collagen-induced arthritis studies in mice [68,70], treatment with FTY720 (0.3 or 0.6 mg/kg) reduced the number of both lymphocytes and monocytes in the blood. Because FTY720 suppresses inflammatory cytokine release in tissues, FTY720 could reduce the chemotaxis of various inflammatory cells (including T cells, dendritic cells, and monocytes) from blood to peripheral tissues.

#### 4.2.3. Role of FTY720 in RANKL-Induced Osteoclastogenesis

We have demonstrated that FTY720 suppressed osteoclastogenesis induced by RANKL in murine bone marrow cells [72]. Treatment with FTY720 decreased osteoclastogenic genes, including Nfactc1, Ctsk, Acp5, and Oscar induced by RANKL, compared with vehicle treatment [72]. Our study suggests that FTY720 modulates cell signaling pathways induced by RANKL. Additionally, reducing the number of infiltrated T cells in the bone and surrounding soft tissues by FTY720 could further decrease RANKL production, subsequently suppressing osteoclastogenesis. In an apical periodontitis animal study, the number of osteoclasts and RANKL expression decreased significantly in the periodontal tissues of the FTY720-treated group compared with the control group [31]. In summary, studies from our and other labs support that FTY720 attenuates inflammatory bone loss by reducing inflammatory cytokine release, inhibiting inflammatory cell migration from blood to bone and surrounding soft tissues, and suppressing osteoclastogenesis induced by RANKL (Figure 3, Table 2). 

## 5. Conclusions

Our understanding of the role of S1P and S1PRs in osteoimmunology continues to evolve rapidly. New discoveries show that S1PRs play multifaceted roles in regulating immune responses, which cannot be explained simply by explicating S1P’s signaling pathway. Studies by our team and other groups have demonstrated that S1PRs, as G protein coupled receptors, control the S1P signaling pathway and regulate cellular signaling pathways induced by other stimuli, including bacteria, bile acid, cytokines (such as IL-6 and RANKL), and hormones (such as vitamin D). Notably, pharmacological inhibition of S1PR2 by JTE013, or treatment with the S1PRs modulator (FTY720) suppressed inflammatory bone loss by reducing inflammatory cytokine release, inhibiting monocyte chemotaxis, and repressing RANKL-induced osteoclastogenesis.

Although JTE013 and FTY720 inhibited inflammatory bone loss in animals, there are some challenges to using them in humans. First, these drugs can generate potential side effects. FTY720 can cause lymphopenia [77,78,79], which impairs immune response against bacterial or viral infection. Treatment with JTE013 exhibited both beneficial effects and side effects associated with tumor growth, chemotherapy, and metastasis [80,81,82,83,84]. Treatment with JTE013 enhanced tumor cell migration, which can potentially lead to tumor invasion and metastasis [80,81]. In contrast, other studies demonstrated that treatment with JTE013 was able to inhibit the proliferation of tumor cells in vitro [82] and in vivo [81]. Treatment with JTE013 also improved chemotherapy efficacy against Ewing sarcoma [83] and sensitized 5-fluorouracil chemotherapy against colorectal cancer [84]. Second, JTE013 might be unstable in vivo and is rapidly metabolized in the liver [81]. Therefore, there is a need to develop more stable S1PR modulators to treat inflammatory bone loss disease. Additionally, using tissue-engineering techniques to conjugate these drugs in biodegradable polymers might be a good strategy to treat inflammatory bone loss diseases in humans. Local delivery of these drug-conjugated polymers in tissues allows slow release of the drug in disease-affected tissues while eliminating potential side effects associated with systemic administration of the drug.

## Figures and Tables

**Figure 1 ijms-22-04411-f001:**
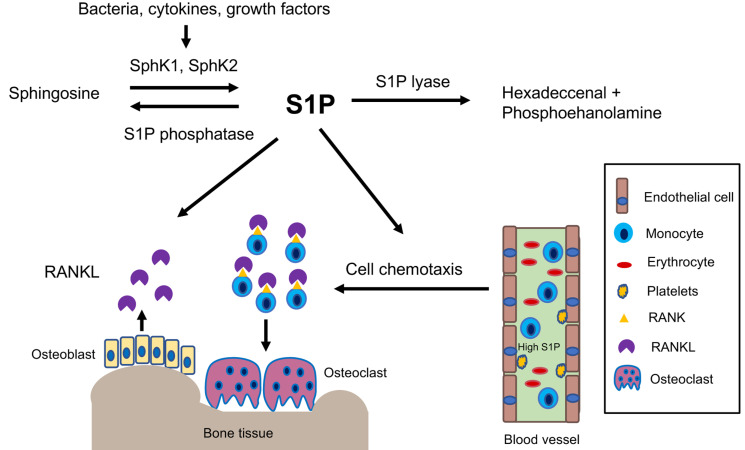
Simplified illustration of biological effects of S1P in inflammatory bone loss diseases. S1P is generated by activation of sphingosine kinases. S1P can be irreversibly degraded by S1P lyase or can be dephosphorylated by S1P phosphatase. S1P acts mainly as a chemokine to promote the chemotaxis of osteoclast precursors from blood circulation to bone and soft tissues. Additionally, S1P can induce RANKL production, which binds with RANK initiating osteoclastogenesis.

**Figure 2 ijms-22-04411-f002:**
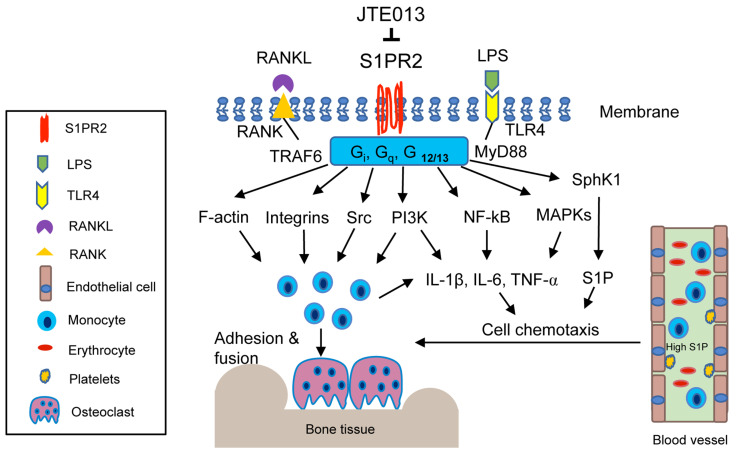
Simplified illustration of biological effects of S1PR2 in inflammatory bone loss diseases. S1PR2 couples with G_i_, G_q_, and G_12/13_ proteins, which manipulate multiple signaling pathways affecting inflammatory diseases. First, S1PR2 controls PI3K, NF-κB, and MAPKs signaling pathways induced by LPS, influencing IL-1β, IL-6, TNF-α inflammatory cytokine release. Second, S1PR2 modulates Sphk1 and S1P generation induced by inflammation. Third, S1PR2 manipulates chemotaxis of osteoclast precursors from blood circulation to bone and soft tissues by modulating inflammatory cytokine and S1P generation. Fourth, S1PR2 controls podosome components (F-actin, integrins, Src, PI3K) induced by RANKL and modulates the adhesion and fusion of osteoclast precursors. S1PR2 possibly interacts with G_i_, G_q_, and G_12/13_ proteins, RANKL adaptor protein TRAF6, and TLR4 adaptor protein MyD88 in lipid rafts after stimulation by RANKL or LPS.

**Figure 3 ijms-22-04411-f003:**
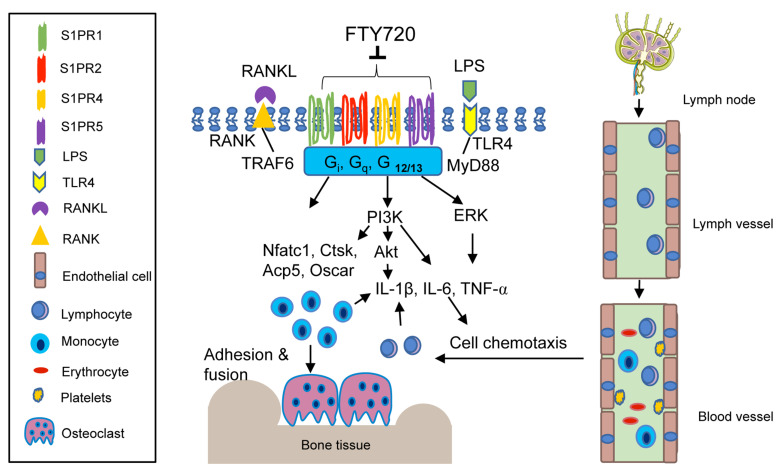
Simplified illustration of biological effect of FTY720 in inflammatory bone loss diseases. FTY720 can internalize and partially degrade multiple S1PRs, including S1PR1, S1PR2, S1PR4, and S1PR5. FTY720 inhibits IL-1β, IL-6, and TNF-α inflammatory cytokine release by blocking PI3K, Akt, and ERK signaling pathways. FTY720 suppresses T lymphocyte egress from draining lymph nodes to the lymph and peripheral blood. Reduction of inflammatory cytokines and the number of T lymphocytes in the blood subsequently decreases monocytes and T lymphocytes in the bone and soft tissues. Additionally, FTY720 inhibits Nfact1, Ctsk, Acp5, and Oscar genes, suppressing RANKL-induced osteoclastogenesis.

**Table 1 ijms-22-04411-t001:** Biological effects of S1PR2 in response to various stimuli.

Effects	Signal Pathways & Mechanisms	Ref.
Knockdown of S1PR2 by S1PR2 shRNA or inhibition of S1PR2 by JTE013 in murine BMMs reduced IL-1β, IL-6, and TNF-α levels induced by *A. actinomycetemcomitans*	PI3K, NF-κB, MAPKs	[42,43]
S1PR2 deficiency or inhibition of S1PR2 by JTE013 reduced bacterial burden and improved survival rate in mice with sepsis	Rac, F-actin	[48]
S1PR2 deficiency or inhibition of S1PR2 by JTE013 suppressed IL-1β and IL-18 induced by LPS in plasma	Unclear	[41]
Treatment with S1PR2 siRNA affected monocyte migration induced by S1P	PI3K, Rac	[39,49]
Inhibition of S1PR2 by JTE013 reduced IL-1β and attenuated colitis induced by both deoxycholic acid and dextran sulfate sodium	ERK, cathepsin B	[45]
Knockdown of S1PR2 by S1PR2 shRNA or inhibition of S1PR2 by JTE013 in murine bone marrow cells suppressed osteoclastogenesis and bone resorption induced by RANKL	Podosome components (PI3K, Src, Pyk2, F-actin, integrins, and paxillin)	[42,43]
Inhibition of S1PR2 by JTE013 decreased IL-4, IL-5, and IL-13 in bronchoalveolar lavage fluid and attenuated inflammation in the lungs of mice induced by ovalbumin	NF-κB	[46]
*S1pr2*^−/−^ mice reduced IL-13, IL-4, CCL17, and CCL24 in the bronchoalveolar lavage fluid and inhibited lung fibrosis induced by bleomycin	STAT6	[47]
Vitamin D and its analog reduced S1PR2 mRNA levels in monocytes and alleviated ovariectomy-induced osteoporosis	Unclear	[58]
An IL-6 receptor antibody decreased S1PR2 mRNA levels in monocytes and alleviated collagen-induced bone loss	Unclear	[59]
Treatment with S1PR2 siRNA or JTE013 in mice alleviated liver inflammation and fibrosis induced by bile duct ligation	NLRP3 inflammasome	[44,49]
Inhibition of S1PR2 by JTE013 reduced osteoporosis induced by RANKL	Cell migration from blood to bone tissues	[39]
Inhibition of S1PR2 by JTE013 attenuated periodontal inflammation and alveolar bone loss induced by tooth ligature placement	PI3K, NF-κB, MAPKs	[40]

**Table 2 ijms-22-04411-t002:** Biological effects of FTY720 in regulating inflammatory bone loss response.

Effects	Signal Pathways & Mechanisms	Ref.
Reduced IL-1β, IL-6, and TNF-α levels induced by *A. actinomycetemcomitans* in murine BMMs	PI3K, Akt, ERK	[72]
Suppressed inflammatory cytokine production Induced by LPS	Altered surface markers and antigen presentation	[73,74]
Shifted cytokine profile from Th1 to Th2 in dendritic cells	Unclear	[64]
Induced T cell apoptosis	Increased cell membrane permeability	[75]
Decreased the number of T cells in lymph andblood	Degraded S1PR1, which regulates the egress of T lymphocytes from secondary lymphoid organs to lymph and blood	[70,71,76,77,78,79]
Inhibited dendritic cell chemotaxis in response to chemokines	Decreased F-actin	[64]
Suppressed osteoclastogenesis induced by RANKL	Reduced osteoclastogenic genes, including Nfatc1, Ctsk, Acp5, and Oscar	[72]
Inhibited arthritis induced by either adjuvant or collagen	Lymphopenia, reduced inflammatory cytokine production, Inhibited generation of collagen antibody, suppressed Dendritic cell migration to draining lymph nodes	[67,68,69,70,71]
Attenuated apical periodontitis	Decreased RANKL production, inhibited osteoclastogenesis	[31]
Alleviated ovariectomy-induced osteoporosis	Reduced monocyte migration from blood to bone surface	[66,67]

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
