# Peer review of "Targeting S1PRs as a Therapeutic Strategy for Inflammatory Bone Loss Diseases—Beyond Regulating S1P Signaling"

_ijms, 2021, doi:10.3390/ijms22094411_

Round 1

Reviewer 1 Report

The review presented by Yu covers the current and recent studies about S1PRs as potential therapeutic targets for inflammatory bone loss diseases. The author focused in this review in summarizing the different mechanisms associated with the SIPRs ligands (mainly, JTE013 and FTY720) in regulating the inflammatory cytokine release, cell chemotaxis, and osteoclastogenesis, and their implications in influencing inflammatory bone loss diseases.

The presented review covers a hot research topic. The review is well presented and covers the most recent developments and studies about S1PRs as potential therapeutic target in inflammatory bone loss diseases.

I have some suggestions to the author:

  • why the author just focused about JTE013 and FTY 720? although there are other derivatives for these compounds which showed to be better metabolized and showed significant potency.
  • extend the introduction part. It is short and does not fit to the topic. The author mainly discussed/introduced RANKL? I would suggest that author should introduce S1P, S1PRs and their binders FTY720 and JTE013.
  • I think to include the effect of S1PRs and their modulator on the cancer related bone metastasis would be valuable in this review.
  • it would be interesting if the author add/discuss about the effect of S1PRs on the crosstalk between osteoblasts and osteoclasts.
  • table 1 does not fully covers the different biological effects of FTY720 in regulating immune responses, please check it and modify it.
  • I suggest that the author summerize the biological effects of JTE013 modulator in a table as for FTY720
  • please cite figure 2 and 3 in the text.
  • I can not find Figure 3? please included.
  • in the conclusion part, please add future prospective/direction in this topic and how could be the potency of discovering new S1PRs modulators and drug discovery for inflammatory bone diseases.
  • please, carefully check MS, there are some typo mistakes.

Author Response

The review presented by Yu covers the current and recent studies about S1PRs as potential therapeutic targets for inflammatory bone loss diseases. The author focused in this review in summarizing the different mechanisms associated with the SIPRs ligands (mainly, JTE013 and FTY720) in regulating the inflammatory cytokine release, cell chemotaxis, and osteoclastogenesis, and their implications in influencing inflammatory bone loss diseases.

The presented review covers a hot research topic. The review is well presented and covers the most recent developments and studies about S1PRs as potential therapeutic target in inflammatory bone loss diseases.

I have some suggestions to the author:

  1. Why the author just focused about JTE013 and FTY 720? although there are other derivatives for these compounds which showed to be better metabolized and showed significant potency.

Response: We previously have used W146 (S1PR1 antagonist), JTE013 (S1PR2 antagonist), Cay 10444 (S1PR3 antagonist), and FTY720.  Only JTE013 and FTY720 reduced inflammatory cytokine IL-1b, IL-6, and TNF cytokine expression induced by the oral pathogen A. actimycetemcomitans. Additionally, JTE013 and FTY720 inhibited RANKL-induced osteoclastogenesis. We did not aware other compounds that are better metabolized and have significant potency compared with JTE013 and FTY720.

  1. Extend the introduction part. It is short and does not fit to the topic. The author mainly discussed/introduced RANKL? I would suggest that author should introduce S1P, S1PRs and their binders FTY720 and JTE013. 

Response: Thank you for the suggestion. In the revised introduction part, I included S1P, S1PRs, JTE013 and FTY720 from page 3 line 60 to page 4 line 69.

  1. I think to include the effect of S1PRs and their modulator on the cancer related bone metastasis would be valuable in this review.

Response:  The effects of S1PRs on cancer related bone metastasis have been presented by Lincheng Zhang et al. from page 4396 to 4397 in this review: Sphingosine-1-phosphate (S1P) receptors: Promising drug targets for treating bone-related diseases.  J Cell Mol Med. 2020 Apr;24(8):4389-4401.  Because I did not find additional new manuscripts about this topic, I decide no to repeat this part.

  1. It would be interesting if the author add/discuss about the effect of S1PRs on the crosstalk between osteoblasts and osteoclasts.

Response: The effect of S1PRs on the osteoblasts and the effect of S1PRs on the crosstalk between osteoblasts and osteoclasts have been presented by Lincheng Zhang et al. from page 4391 to 4392 in this review: Sphingosine-1-phosphate (S1P) receptors: Promising drug targets for treating bone-related diseases.  J Cell Mol Med. 2020 Apr;24(8):4389-4401. Because I did not find additional manuscripts related to the crosstalk between osteoclasts and osteoblasts, I decide no to repeat this part.  We just want to emphasize the roles of S1PRs in regulating various of stimuli beyond S1P signaling, which has not been mentioned in other review manuscripts.

  1. Table 2 does not fully covers the different biological effects of FTY720 in regulating immune responses, please check it and modify it.

      Response: Thanks for the suggestion. I changed the tile of Table 2 as “ Biological effects of FTY720 in regulating inflammatory bone loss response”.

  1. I suggest that the author summerize the biological effects of JTE013 modulator in a table as for FTY720

      Response: Thanks for the suggestion. I changed the Table 1 format as Table 2 format.

  1. Please cite figure 2 and 3 in the text.

      Response: Thanks for the suggestion. I cited the figure 2 and figure 3 in the text of revised manuscript.

  1. I cannot find Figure 3? please included.

      Response: The IJMS journal website only allows me to upload maximum two figures. I have to upload the figure 3 as a supplement. Please check the supplement figure. 

  1. In the conclusion part, please add future prospective/direction in this topic and how could be the potency of discovering new S1PRs modulators and drug discovery for inflammatory bone diseases.

Response: Thanks for the suggestion. In the revised conclusion part, I discussed the side effects associated with using JTE013 or FTY720, and future direction to use tissue-engineering techniques to deliver S1PRs modulator locally in the tissues (line 398 to line 413).

  1. please, carefully check MS, there are some typo mistakes.

Response: I select ijms English service for correcting any English errors.

Reviewer 2 Report

The manuscript covers the molecular mechanisms of S1PR related signaling in inflammatory diseases. In particular, the authors focus on the S1P signaling stimulated by some kinds of factors other than S1P itself. The focus is clear but the following minor points should be reconsidered to help the readers to understand the contents of this manuscript and future strategy of this field.

  1. Introduction. The authors should point out the effects of inflammatory factors on osteoblast function as well as the osteoclasts dysfunction focusing on the S1P-mediated osteoblast-osteoclast coupling. For example, inflammatory responses or cancer progression deteriorate the bone function by triggering the osteoblasts dysfunction (Fir example, International Journal of Molecular Sciences, 21, (2020), 6659, Biomolecules, (2021), 11 131.).
  2. Why the authors focused on S1PR2? S1PR2 show specific function mediating cellular migration, leading to modulate the cancer metastasis. the additional description about the specificity of S1PR2 on  cellular function with citation of some references (For example, Nature 541, 233–236 (2017) is necessary.
  3. Some major reported findings should be joined to the figures. The present figures only illustrate the S1P and the related molecules in signaling cascade. More specific detail figures linking S1PR signaling and cellular functions from the referring articles can satisfy the readers of this journal.
  4. FTR720 is a powerful therapeutic target in treatment of inflammatory diseases. The limitation of this drug in treatment of bone disease should also be mentioned.

Author Response

The manuscript covers the molecular mechanisms of S1PR related signaling in inflammatory diseases. In particular, the authors focus on the S1P signaling stimulated by some kinds of factors other than S1P itself. The focus is clear but the following minor points should be reconsidered to help the readers to understand the contents of this manuscript and future strategy of this field.

  • Introduction. The authors should point out the effects of inflammatory factors on osteoblast function as well as the osteoclasts dysfunction focusing on the S1P-mediated osteoblast-osteoclast coupling. For example, inflammatory responses or cancer progression deteriorate the bone function by triggering the osteoblasts dysfunction (Fir example, International Journal of Molecular Sciences, 21, (2020), 6659, Biomolecules, (2021), 11 131.).

Response: Thanks for the suggestion. In the revised introduction (line 58-60), I mentioned that inflammatory cytokines inhibit osteoblast proliferation and disrupt unidirection osteoblast alignment. We cited the two manuscripts that reviewer’s suggested (Ref 14, 15).

  • Why the authors focused on S1PR2? S1PR2 show specific function mediating cellular migration, leading to modulate the cancer metastasis. the additional description about the specificity of S1PR2 on  cellular function with citation of some references (For example, Nature 541, 233–236 (2017) is necessary.

Response: Because my lab focus on studying periodontal disease. We previously have used W146 (S1PR1 antagonist), JTE013 (S1PR2 antagonist), Cay 10444 (S1PR3 antagonist), and FTY720.  Only JTE013 and FTY720 reduced inflammatory cytokine IL-1b, IL-6, and TNF cytokine expression induced by the oral pathogen A. actimycetemcomitans. Additionally, JTE013 and FTY720 inhibited RANKL-induced osteoclastogenesis. Because FTY720 cause lymphopenia side effect, I focused on S1PR2 signaling in regulating periodontal disease.

      Previous studies using JTE013 have shown both beneficial and side effects associated with tumor growth, metastasis, and chemotherapy. In the revised discussion part, I discussed both beneficial and side effects of using JTE013 in tumor studies (line 398 to line 413).

       I searched the manuscript “Genome-wide in vivo screen identifies novel host regulators of metastatic colonization”, which published on Nature 2017  Jan 12;541(7636):233-236. This manuscript focused on the role of S1P transporter spinster homologue 2 (Spns2)-deficient mice on pulmonary tumor metastasis.  It did not mention S1PR2.  I do not understand why the reviewer asked me to cite this manuscript.

  • Some major reported findings should be joined to the figures. The present figures only illustrate the S1P and the related molecules in signaling cascade. More specific detail figures linking S1PR signaling and cellular functions from the referring articles can satisfy the readers of this journal.

Response: S1PRs have many functions. In the figures, we just want to emphasize how S1PRs regulate inflammatory cytokine production, cell chemotaxis, and osteoclastogenesis. I do not know which manuscript figure format that the reviewer recommend to use.

  • FTR720 is a powerful therapeutic target in treatment of inflammatory diseases. The limitation of this drug in treatment of bone disease should also be mentioned.

Response: Thanks for the suggestion. In the conclusion part (line 398 to line 410), we discuss the side effects associated with using FTY720 and JTE013, and discuss the potential to use tissue-engineering technique to deliver the drug locally in the tissues to avoid side effect associated with systemic administration of drugs.

Round 2

Reviewer 1 Report

Many thanks for the author for addressing all points that have been raised. The review has significantly improved. I would definitely recommend the publication of this very interesting and important review.